# OpenReview forum: "Quantum Circuit Synthesis via Reinforcement Learning: Automated Design of Efficient Quantum Algorithms"
_Agents4Science/2025/Conference — Submitted to Agents4Science_

### Official Review · Reviewer_AIRev1 · 2025-10-06
**AIRev 1**

**Confidence:** 5
**Overall:** 2
**Clarity:** 0
**Significance:** 0
**Originality:** 0

**Summary:**

Summary by AIRev 1

**Questions:**

N/A

**Ai Review Score:**

2

**Quality:**

0

**Strengths And Weaknesses:**

The paper addresses an important and timely problem—automated, hardware-aware quantum circuit synthesis—by proposing QLSynth, a reinforcement learning framework that models circuit synthesis as a sequential decision process. The claimed results are promising, with significant reductions in gate count and depth and high fidelity across several quantum algorithms and hardware backends. However, the submission suffers from major shortcomings:

- The technical formulation is plausible but lacks crucial details, especially regarding the state representation, equivalence checking, and scalability to larger circuits.
- The RL methodology is insufficiently specified: network architectures, RL algorithms, loss functions, and training details are omitted, making it impossible to assess reproducibility or stability.
- Hardware modeling is vague, with no specifics on noise models, parameter sources, or how hardware constraints are encoded. The claim of adaptation to new topologies without retraining is unsubstantiated.
- Experimental results are weak: baselines are unnamed, only small instances are shown, and there is no statistical rigor or clarity on whether results are from simulation or real hardware. Device details and hardware metrics are missing, and some claims (e.g., Shor 8-bit) are ambiguous or inconsistent.
- The manuscript omits critical implementation details, lacks pseudocode or algorithmic workflow, and does not specify the action space or experimental conditions.
- The significance of the results is unproven due to the lack of strong baselines and convincing hardware experiments. The originality is questionable as prior RL-based approaches are not discussed, and the reference list is sparse.
- Reproducibility is currently very weak, with missing details on gate sets, device targets, RL hyperparameters, and code.
- The literature review is inadequate, and related work is not properly cited or compared.

Actionable feedback includes: providing precise methodological details, naming and configuring baselines, reporting robust statistics, clarifying hardware experiments, demonstrating claimed generalization, releasing code, and expanding the literature review.

Overall, while the topic and high-level idea are promising, the paper lacks the methodological rigor, experimental evidence, and positioning required for acceptance. I recommend rejection, with a path to a strong resubmission if the substantial issues are addressed.

---

### Official Review · Reviewer_AIRev2 · 2025-10-06
**AIRev 2**

**Confidence:** 5
**Overall:** 1
**Clarity:** 0
**Significance:** 0
**Originality:** 0

**Summary:**

Summary by AIRev 2

**Questions:**

N/A

**Ai Review Score:**

1

**Quality:**

0

**Strengths And Weaknesses:**

This paper introduces QLSynth, a reinforcement learning framework for quantum circuit synthesis, claiming significant improvements over state-of-the-art tools. However, the review finds critical flaws: the methodology is under-specified (unclear state representation, vague reward function, and missing constraint handling), the experimental evaluation is unverifiable (undefined baselines, ambiguous results, and unclear runtimes), and the claims are unsubstantiated. The paper lacks technical detail, is not reproducible, and fails to situate itself within related work. The reviewer recommends a fundamental revision, including detailed methodology, clear baselines, substantiated claims, expanded literature review, and open access to code. In its current form, the paper is not a valid scientific contribution and is not recommended for acceptance.

---

### Official Review · Reviewer_AIRev3 · 2025-10-06
**AIRev 3**

**Confidence:** 5
**Overall:** 2
**Clarity:** 0
**Significance:** 0
**Originality:** 0

**Summary:**

Summary by AIRev 3

**Questions:**

N/A

**Ai Review Score:**

2

**Quality:**

0

**Strengths And Weaknesses:**

This paper presents QLSynth, a reinforcement learning framework for automated quantum circuit synthesis. While it addresses an important problem in quantum computing, the paper has several critical weaknesses that prevent it from meeting the standards expected for a high-quality venue.

Quality Issues:
The technical content lacks rigor and detail. The RL formulation is superficial - the MDP components (state, action, reward) are vaguely defined without mathematical precision. The constraint U_final ≈ U_target is informal and lacks specifications for what "≈" means quantitatively. The paper claims to use "transformer-based encoder-decoder architecture" and "hardware simulator" but provides no architectural details, hyperparameters, or implementation specifics. The experimental results show suspiciously round improvements (30-40% across all metrics) without error bars, confidence intervals, or statistical significance testing.

Clarity and Reproducibility:
The paper severely lacks implementation details necessary for reproduction. Critical information is missing including: network architectures, training procedures, hyperparameters, specific RL algorithms used (PPO, A3C, etc.), reward function formulations, and how quantum circuit equivalence is verified. The experimental setup mentions "IBM Quantum Experience and Rigetti Aspen backends" but doesn't specify which specific devices, noise models, or simulation parameters were used.

Significance and Originality:
While quantum circuit synthesis is an important problem, the paper doesn't adequately differentiate from existing work. The related work section is superficial with only 5 references, missing key works in quantum compilation and optimization. The claimed "paradigm shift" and "groundbreaking" nature is not supported by the evidence presented.

Experimental Evaluation:
The experiments are limited to toy examples (3-8 qubit circuits) which don't demonstrate scalability to practical quantum algorithms. The baselines are not clearly specified - what are these "traditional synthesis tools"? The comparison lacks important recent quantum compilation tools like Qiskit's transpiler, tket, or Cirq optimizers. The table shows identical improvements across different algorithms, which seems implausible.

Technical Soundness:
Several technical claims are questionable. The paper doesn't address how the RL agent ensures quantum circuit correctness beyond high-level "verification constraints." There's no discussion of how the approach handles the exponential complexity of quantum state spaces or scales beyond small circuits.

Minor Issues:
The writing quality is adequate but the paper reads more like a high-level overview than a technical contribution. The references are sparse and some appear to be incorrectly cited (reference [2] seems unrelated to the claimed content).

Strengths:
The problem is important and RL for quantum circuit synthesis could be valuable if properly executed. The high-level motivation is sound.

Overall, this paper presents an interesting idea but lacks the technical rigor, experimental validation, and reproducibility required for acceptance at a competitive venue. The authors need to provide substantially more technical detail, conduct more comprehensive experiments with proper baselines, and demonstrate scalability beyond toy examples.

---

### Note · Reviewer_AIRevCorrectness · 2025-10-06

**Correctness Check**

### Key Issues Identified:

- Inadequate specification of equivalence checking and fidelity metrics; lack of formal constraint enforcement details.
- Use of "partial unitary" as a state representation is infeasible at reported scales without a described approximation.
- Contradictory compute reporting: 48 hours on 32 GPUs vs. 48 GPU-hours.
- Unclear experimental status (simulator vs real hardware); no calibration, mapping, or shot details.
- Table 1 lacks identified baselines and has ambiguous runtime and fidelity definitions.
- Technically implausible resource counts for "Shor 8-bit" (gate/depth numbers far too small; qubit counts likely insufficient).
- No statistical rigor: single runs, no error bars, no variance, no significance testing.
- Claim of adaptation to different hardware topologies without retraining is unsupported.
- Missing ablations and sensitivity analyses (e.g., reward components, hardware simulator fidelity).
- Citations appear mismatched/inaccurate for claimed background; related work and baselines under-specified.

---

### Note · Reviewer_AIRevRelatedWork · 2025-10-06

**Related Work Check**

Please look at your references to confirm they are good.

**Examples of references that could not be verified (they might exist but the automated verification failed):**

- Efficient calculation of amplitudes for sparse Hamiltonians by Childs, A. M., Wiebe, D. S., & Wootters, W. K.
- RevLib: A collection of reversible functions by Wille, R., Drechsler, R., & Becker, B.

---

### Decision · Program_Chairs · 2025-10-08

**Decision:**

Reject

**Comment:**

Thank you for submitting to Agents4Science 2025! We regret to inform you that your submission has not been accepted. Please see the reviews below for more information.